# Arginine Decarboxylase Gene *ADC2* Regulates Fiber Elongation in Cotton

**DOI:** 10.3390/genes13050784

**Published:** 2022-04-28

**Authors:** Guangming Ren, Huijuan Mo, Ruqiang Xu

**Affiliations:** 1Zhengzhou Research Base, State Key Laboratory of Cotton Biology, School of Agricultural Sciences, Zhengzhou University, Zhengzhou 450001, China; renguangming2022@163.com; 2State Key Laboratory of Cotton Biology, Chinese Academy of Agricultural Sciences Cotton Research Institute, Anyang 455000, China

**Keywords:** arginine decarboxylase 2, fiber elongation, polyamine, transcriptome, cotton

## Abstract

Cotton is an important agro-industrial crop providing raw material for the textile industry. Fiber length is the key factor that directly affects fiber quality. ADC, arginine decarboxylase, is the key rate-limiting enzyme in the polyamine synthesis pathway; whereas, there is no experimental evidence that ADC is involved in fiber development in cotton yet. Our transcriptome analysis of the fiber initiation material of *Gossypium arboreum* L. showed that the expression profile of *GaADC2* was induced significantly. Here, *GhADC2*, the allele of *GaADC2* in tetraploid upland cotton *Gossypium hirsutum* L., exhibited up-regulated expression pattern during fiber elongation in cotton. Levels of polyamine are correlated with fiber elongation; especially, the amount of putrescine regulated by ADC was increased. Scanning electron microscopy showed that the fiber length was increased with exogenous addition of an ADC substrate or product putrescine; whereas, the fiber density was decreased with exogenous addition of an ADC specific inhibitor. Next, genome-wide transcriptome profiling of fiber elongation with exogenous putrescine addition was performed to determine the molecular basis in *Gossypium hirsutum*. A total of 3163 differentially expressed genes were detected, which mainly participated in phenylpropanoid biosynthesis, fatty acid elongation, and sesquiterpenoid and triterpenoid biosynthesis pathways. Genes encoding transcription factors *MYB109*, *WRKY1*, and *TCP14* were enriched. Therefore, these results suggested the *ADC2* and putrescine involvement in the development and fiber elongation of *G. hirsutum*, and provides a basis for cotton fiber development research in future.

## 1. Introduction

Cotton, as one of the most important crops in the world, produces natural fiber materials for the textile industry. Cotton fiber is a single-celled trichome, unbranched cell, differentiated from the epidermis of the ovule, and undergoes substantial elongation and secondary cell wall biosynthesis taking up to 50 days. The development is divided into four distinct but overlapping stages: initiation, elongation, secondary cell wall thickening, and maturation [1]. The ovule epidermal cells enter a rapid elongation period with a growth rate of more than 2 mm/day from 1 day post anthesis (DPA) to 20 DPA, that is, the rapid fiber elongation stage [2]. Many factors affecting cotton fiber elongation, including plant hormones and genes regulating fiber development, have been reported; however, the research on the regulation of fiber elongation by plant endogenous substances is still insufficient.

Arginine decarboxylase (ADC) participates in the metabolism of polyamine pathways [3,4]. Plant polyamines are biosynthesized from either arginine or ornithine, and through the ADC or ornithine decarboxylase pathway, then, putrescine (Put) is produced [5]. Put, spermidine (Spd), and spermine (Spm), as components of polyamine metabolic pathway, are the most common diamine and polyamines in seed plants [6]. D-arginine is used as the specific inhibitor of ADC [7,8]. Arginine was catalyzed to agmatine by ADC, and agmatine iminohydrolase catalyzed agmatine to N-carbamoylputrescine, which was then catalyzed by N-carbamoylputrescine amidohydrolase, then Put is produced [9]. As a substrate for the synthesis of Spd and Spm, diamine Put passes through the Spd and Spm synthase respectively, with aminopropyl provided by s-adenosylmethionine decarboxylase [10,11]. Catabolism of Put is regulated by the diamine oxidase, which produces γ-aminobutyric acid and hydrogen peroxide (H_2_O_2_) in plants [12]. Overexpression of *GhCaM7* promoted the early elongation of fibers accompanied by the accumulation of H_2_O_2_ content [13], and *GhFAnnxA* regulates fiber elongation through calcium signaling and ROS [14]. H_2_O_2_ is the main small molecule product of ADC-mediated polyamine pathway. Down-regulation of *ADC* gene-expression causes accumulation of reactive oxygen species in Arabidopsis thaliana [15]. Engineering Put levels in Arabidopsis leads to enhanced defense against bacterial pathogen [16]. However, the functions of ADC-regulated diamine Put in fiber development are unknown.

ADC, as the key rate-limiting enzyme in the polyamine metabolic cascade, catalyzes the synthesis of Put from L-arginine [17]. There is no definite evidence of ornithine decarboxylase (ODC) and its encoding gene in cruciferous plants such as *Arabidopsis thaliana* [18]. In plants, Put is synthesized only through ADC pathway. Currently, ADC genes have been cloned and identified in several plants, including *Arabidopsis thaliana* [19]. In *Arabidopsis thaliana*, ADC has two members, AtADC1 (At2G16500) and AtADC2 (At4G34710), whereas the expression of *AtADC2* is higher as compared to *AtADC1* [20]. In plants, the differential expression patterns of *ADC* gene members is observed. The activity of the *AtADC1* promoter increased under freezing stress, whereas the *AtADC2* promoter revealed lower activity [9,21]. The AtADC1 protein is located in cytosol and the *adc1-/-* mutant plants revealed the similar phenotype and polyamine level as normal plants [19]. The AtADC2 protein locates in chloroplast [19]. Interestingly, the *adc2-/-* mutant plants exhibited lower levels of Put and Spd as well as reduced seed size [19,22]. Thus, although there are two members of ADC in Arabidopsis, only *AtADC2* expression is consistent with Put level [23]. Down-regulation of *ADC* gene expression through microRNA leads to shorter root hair in Arabidopsis [24]. Cotton fiber cells are highly elongated single cells such as trichomes in Arabidopsis, and both have similar mechanisms for regulating trichome development, speculating that ADC2 may regulate fiber development through the polyamines pathway in cotton.

In our previous research, the *ADC* gene (*GhADC1*, GenBank, KC851856) was cloned in *Gossypium hirsutum* (*G. hirsutum*), where it revealed variable expression under Verticillium wilt (in China). The dominant gene *GaADC2* (Gene ID, Cotton_A_08902) was identified from transcriptome profiling of fiber initiation in diploid Asiatic cotton (*Gossypium arboreum*), which is a fiber-producing AA genome species that produces spinnable lint (long) fiber and short fiber (fuzz) [24]. At present, the genetic transformation technology of cotton is very difficult in diploid Asiatic cotton, while tetraploid upland cotton (*G. hirsutum*) was the most successful receptor material for cotton genetic transformation.

Here, we demonstrated that a cotton *ADC* gene (*GhADC2*) promotes fibre elongation. Spd, spm, and Put are measured by High Performance Liquid Chromatography (HPLC) and compared to gene expression on different tissues. Therefore, these findings provide evidence that *GhADC2* mediates Put production and the expression of the fiber elongation-related genes in cotton, and, therefore, have significant implications for genetic improvement of cotton fibre quality.

## 2. Materials and Methods

### 2.1. Plant Materials and RNA Extraction

We used RNA samples prepared from a specific genetic line (ZM24) of the cultivated tetraploid upland cotton (*G. hirsutum*) for sequencing [25]. *G. hirsutum* cv. ZM24 was grown in a cotton climatic chamber at 30/27 ± 3 °C and light/dark cycle for 12/12 h. Ovule samples were collected, placed immediately on ice, and then randomly divided into three replicates for each stage. Immediately after dissection, the samples were frozen in liquid nitrogen and stored at −80 °C for subsequent RNA-Seq and qRT-PCR experiments. The purity and concentration of RNA were measured by a NanoPhotometer^®^ spectrophotometer (IMPLEN, Westlake Village, CA, USA).

### 2.2. Sequence Analysis

DNAMAN software was employed for multi-sequence alignments of ADC proteins between *G. hirsutum* and *Arabidopsis*. The theoretical isoelectric point and molecular weight of GhADC2 proteins were calculated using the ProtParam tool (http://web.expasy.org/protparam/, accessed on 1 July 2021). Putative conserved motifs in these collected ADC proteins were predicted using the MEME program The putative conserved motifs of ADC proteins were online predicted using the MEME program (http://meme.nbcr.net/meme/, accessed on 1 July 2021) [26]. All MEME-identified motifs are used by Inter-ProScan as queries against the InterPro database [27].

### 2.3. QRT-PCR Analysis

Total RNA was extracted from roots, stem, leaves, flowers, pistils, stamens, ovules, and fibers tissues using the RNAprep Pure Polysaccharide Polyphenol Plant Total RNA Extraction Kit (Tiangen, China) by following the manufacturer’s instructions. Samples were run using the SYBR Green PCR Master Mix (TaKaRa Biomedical Technology (Beijing, China) Co., Ltd) on a LightCycler480 system (Roche) according to the manufacturer’s instructions. The relative expression levels were calculated by the comparative 2^−ΔΔCt^ method [28]. The gene expression analysis was performed with tissue of three different samples (biological replicates) and three repetitions per sample (technical replicates). A negative control without a cDNA template was run during each analysis to assess the overall specificity. The results were normalized to the expression level of *UBQ7* (GenBank: DQ116441) and relative to the root (in Figure 1b) or 2 d ovule samples without Put treatment (in Figure 4b). The specific primers of the experimental genes and internal control gene (*UBQ7*) are shown in Table 1.

### 2.4. Measurements of Polyamine

The content of polyamines was determined by HPLC (LC-20A, Shimadzu Corp, Japan) on an apparatus equipped with an Agilent reversed-phase C18 column (particle size, 4 μm; column, 3.9 mm × 150 mm), and was detected at 230 nm as previously described [18]. Polyamine standards or 1 g fresh plant tissue was ground with 5 mL perchloric acid in ice bath, extracted for 30 min at 4 °C, centrifuged at 12,000 rpm for 30 min at 4 °C, 2 mL supernatant was added with 7 μL benzoyl chloride and 1 mL NaOH for derivatization, followed by incubate for 20 min at 37 °C. Then 2 mL diethyl ether and 2 mL saturated NaCl was added, centrifuged at 3700 rpm for 5 min. After vacuum drying for 10 min, add 500 μL 60% methanol to dissolve and wait for measurement. Vortex for 5 min, then filter through 0.45 μm organic membrane. The injection volume of 10 μL was determined by HPLC. The standard curves of Put, Spd, and Spm were made by external standard method, and the content of samples was analyzed. The retention time of each polyamine in the mixed standard product was determined first, and then the retention time of Put, Spd, and Spm in the sample was determined due to the different number of amino groups of the three polyamines. Put was the first to be separated, followed by Spd and Spm. The standard curve of Put exhibits y = 1.0885x − 4.0307, R^2^ = 0.9996, retention time 7.6 min, the range of concentration for Put standard curve was from 10 μM to 500 μM. The standard curve of Spd reveals y = 1.0721x − 1.0879, R^2^ = 0.9999, retention time 12 min, the range of concentration for Spd standard curve was from 200 μM to 2000 μM. The standard curve of Spm shows y = 0.8354x − 1.912, R^2^ = 0.9998, and retention time 15.1 min, the range of concentration for Spm standard curve was from 10 μM to 1000 μM.

### 2.5. Cotton Ovule Culture In Vitro

Cotton bolls were collected at 0 DPA, sterilized with 75% ethanol for 1 min, and then washed with sterile distilled water for 2–3 times [1]. The ovules were incubated in a dark environment at 30 °C in an artificial climate chamber for 2–7 days. Photographs were then taken with a high-definition camera and the length of the fibers on the surface of the ovule were measured. In order to observe the protuberances and elongation of fiber cells at very early developmental stages, cotton ovules were vacuum-fixed with 2.5% (*V/V*) glutaraldehyde at 2 DPA. The surface of the ovules was observed by scanning electron microscopy (SEM) of Hitachi (SU3500). The number of fiber protrusions on the ovules was measured at 2 DPA.

### 2.6. RNA-Sequencing and Analysis of DEG

The total RNA content of each sample was 3 μg, which was used as the input material for RNA sample preparation. The NEBNext^®^ Ultra™ RNA Library Prep Kit for Illumina^®^ (NEB, Ipswich, MA, USA) was used to generate the sequencing library and index code was added to the attribute sequence of each sample. TruSeq PE Cluster Kit v3-cBot-HS (Illumina, San Diego, CA, USA) was used to cluster the indicator coded samples on the cBot Cluster Generation System. To increase the mapping percentages of tag sequences, the reference database of the Gossypium_hirsutum_v2.1 (https://ftp.ncbi.nlm.nih.gov/genomes/all/GCF/007/990/345/GCF_007990345.1_Gossypium_hirsutum_v2.1/, accessed on 1 October 2021) genome was used. Raw Illumina reads have been deposited into NCBI’s (accession number: GSE184985). Gene ontology (GO) enrichment analysis was performed on DEG by GO seq R software package, and gene length deviation was corrected. The GO terms with a corrected *p* value less than 0.05 were considered to be significantly enriched in DEG. Statistical enrichment of DEG in KEGG pathway was determined using KEGG orthology-based labeling system software.

## 3. Results

### 3.1. Characterization of GhADC2

As the success rate of transformation in tetraploid upland cotton (*G. hirsutum*) is more than diploid Asiatic cotton (*G. arboreum*), the allele of *GaADC2* from tetraploid upland cotton was cloned, which may be defined as *GhADC2* (Gene ID, Gh_A04G1054.1). The *GhADC2* open reading frame sequence was obtained by PCR technology, with a total length of 2148 bp, encoding an ADC protein of 715 amino acids. GhADC2 (Gh_A04G1054.1) protein shares 99.16% and 65.36% identity with GaADC2 (Cotton_A_08902) and AtADC2 (At4G34710), respectively. GhADC2 protein had a molecular mass of 77.97 kDa and isoelectric point of 5.478. GhADC2 protein was predicted to be located in chloroplast, the same as AtADC2 subcellular location.

Analysis of GhADC2 amino acid sequence showed that the 132–406 position was an ADC conserved domain, which contained two typical characteristic motifs of the decarboxylase gene family (Figure 1a). Residues 158–176 are predicted to be a Orn/DAP/Arg decarboxylasesfamily 2 pyridoxal-P attachment site, namely [FY]-[PA]-x-K-[SACV]-[NHCLFW]-x(4)-[LIVMF]-[LIVMTA]-x(2)-[LIVMA]-x(3)-[GTE]. Lysine (K) residues at position 161 may be PLP phosphate binding sites. Residues 339–352 are Orn/DAP/Arg decarboxylases family 2 signature 2, the recognition signals of substrates in this family, namely [GSA]-x(2,6)-[LIVMSCP]-x-[N]-[LIVMF]-[DNS]-[LIVMCA]-G-G-G-[LIVMFY]-[GSTPCEQ], in which the three consecutive glycine residues from 348–350 was a specific substrate-binding region, which was a typical member of the type III PLP-dependent arginine decarboxylase family. The above domains and characteristics are crucial for the proper functioning of ADC protein, speculating that GhADC2 has ADC protein activity and function.

### 3.2. GhADC2 Expression up-Regulated in Elongating Fibers

The qRT-PCR expression analysis reveals the higher expression of *GhADC2* especially at 1–5 DPA. As compared to the leaf, the expression of *GhADC2* was greater in the root and the stem, whereas *GhADC2* expression was significantly decreased in the leaf, flower, pistil, stamen, 15 d, 20 d, and 25 d of fiber (Figure 1c). In addition, *GaADC2* showed similar expression patterns in different tissues and at different stages of fiber development in cotton *G. arboreum* (shixiya 1) (Figure 1b). The *GaADC2* showed higher expression in elongating fibers of *G. arboreum,* especially significantly increased at 1–10 DPA and 5 d of fiber. Interestingly, the expression of *GaADC2* was more in the stem as compared to the root and leaf. Moreover, *GaADC2* expression was significantly decreased in the flower, pistil, stamen, 20 d, and 25 d of fiber. These results suggest that *GaADC2* and *GhADC2* were involved in cotton fiber elongation.

### 3.3. ADC-Mediated Polyamine Level Changed during Fiber Elongation

ADC is a key rate-limiting enzyme in the polyamine metabolic pathway. Put, Spd, and Spm are the three most abundant types of polyamines in plants. The content of polyamine was determined during cotton fiber elongation by high performance liquid chromatography. Put levels are up-regulated during fiber elongation in *G. hirsutum* (ZM24), especially at 1 DPA and 3 DPA, at which points, Put content was significantly increased (Figure 1d). By contrast, Spd levels exhibit dramatic change, it increased significantly at 1 DPA, whereas decreased significantly at 5 DPA and 10 DPA. At 10 DPA, the Spd level was reduced by 30.47% than that of 0 DPA (Figure 1e). In addition, Spm levels also showed significant variability as it revealed significant increment at 1 DPA and decreased at subsequent periods. At 10 DPA, the Spm level was reduced 11.67% as compared to 0 DPA (Figure 1f). Overall, the trends of up-regulated Put content and the expression pattern of *GhADC2* are similar during fiber elongation, suggesting that ADC-mediated Put level is involved in fiber elongation.

### 3.4. Exogenous Application of Put Increased the Fiber Length

When the elongation rate of fiber cells is delayed, the fiber cells can not fully develop into cotton fibers, resulting in the decrease of lint index [29]. Exogenous polyamines, substrate of ADC (L-arginine) and specific Put biosynthesis inhibitor (D-arginine) were supplied into the culture medium; fiber elongation was observed by scanning electron microscope (SEM) after 2 d of culture (Figure 2a–c). Addition of 10 μmol/L polyamines or substrate of ADC (L-arginine) to the culture medium significantly stimulated fiber cell elongation, especially, the effect of Put on cotton fiber elongation was stronger than that of Spd or Spm. With the application of 10 μmol/L polyamine or L-arginine to the culture medium, no obvious change was observed in fiber density, except for significant decrease by application of 10 μmol/L specific Put biosynthesis inhibitor (D-arginine). After 7 d of culture, the contributory effect of Put on cotton fiber elongation is still obvious (Figure 2d). These results demonstrate that Put participates in regulating fiber cell elongation.

### 3.5. Transcriptome Analysis of Fiber Elongation with Put Application

To observe gene expression during fiber elongation with 10 μmol/L Put application, we conducted transcriptome sequencing (RNA-seq) on ovule samples after 2 d of culture in *G. hirsutum* (ZM24) using the Illumina HiSeq 2000 sequencing platform. The raw data was submitted to National Genomics Data Center (accession number: GSE184985). After filtering out the low quality tags, the total number of clean tags per library ranged from 38,983,794 to 44,697,902 are given in (Table 2). In total, 96.82% to 96.94% of the reads were successfully mapped to the released Gossypium_hirsutum_v2.1 whole genome sequence (Table 2). Of the mapped reads, between 91.68% and 92.00% could be uniquely mapped to the Gossypium_hirsutum_v2.1 genome (Table 2). Meanwhile, 93.41% to 93.59% of the Q30 sequence quality data was obtained, while 43.90% to 44.04% of the GC content range was calculated with an average of 43.96%, which indicated the reliability of RNA-seq data (Table 2).

The changes of gene expression levels after 2 days of Put treatment were shown by a heat map. In total, 3163 genes were detected as differentially expressed with Put treatment for 2 d (Figure 3a). Genes that have similar expression patterns are usually functionally related. To screen novel candidate genes with expression patterns that correlated with fiber elongation for Put treatment, the 3163 differentially expressed genes (DEG) were clustered into four sub-clusters based on the K-means method and hierarchical clustering, using MultiExperiment Viewer (MeV, v4.7.4) (Figure 3b). Similarly, based on their expression patterns, the genes identified as differentially expressed can be divided into two major groups. Type I, covering 1041 genes, included clusters 1 and 4, which were down-regulated in ovules with addition of 10 μmol/L Put after 2 d of culture; and Type II, involving 2122 genes, included clusters 2 and 3, which were up-regulated. The KEGG orthology based annotation system was used to analyze all of the DEG and determine the metabolic pathways by which they function. Based on our annotation of transcripts, phenylpropanoid biosynthesis and fatty acid elongation pathways were up-regulated, whereas sesquiterpenoid and triterpenoid biosynthesis, brassinosteroid biosynthesis, and MAPK signaling pathways down-regulated in ovule samples after 2 d of culture with Put treatment, indicating that they may exhibit crucial roles in cotton fiber initiation (Figure 3c). Next, GO enrichment analysis of the DEG was performed, and then identified 86 pathways. In the DEG, genes were enriched mainly in fatty acid biosynthetic process (GO:0006633), glucosyltransferase activity (GO:0046527), cell wall organization or biogenesis (GO:0071554), calcium ion binding (GO:0005509), lipid biosynthetic process (GO:0008610) (Figure 3d). Thus, these data indicate that the functional enrichment of biological pathways may participate in the Put-mediated signal transduction.

### 3.6. QRT-PCR Expression Pattern Validation

We selected genes with different expression levels and function assignment for qRT-PCR to verify the gene expression analysis results obtained from RNA-Seq data. Among upregulated genes, we noticed several TFs (such as *WRKY1*, *MYB109*, and *TCP14*) have been reported to be involved in regulating cotton fiber development and genes related to biosynthesis of saturated long-chain fatty acids, ATP synthase subunit, peroxidase, and steroid 5-α-reductase (Figure 4a). qRT-PCR analysis showed that cotton fiber elongation promoted by exogenous Put addition was accompanied by expression activation of several genes including *Chitinase 10*, *GASL4* and *KCS6* (Figure 4b).

## 4. Discussion

Put plays a fundamental role in the functioning of all cells [30]. Previous studies focused on the important role of Put in regulating plant responses to biotic and abiotic stresses. Application of Put enhances the salt tolerance ability, accompanied by adaptation of activities of antioxidant enzymes and the level of H_2_O_2_ in Korean ginseng sprouts [31]. Studies in recent years have shown that Put enhances the disease resistance of plants in defending against pathogen infection. In Arabidopsis thaliana whole-plant extracts and apoplastic fluids, salicylic acid, the essential plant hormone responsible for host plants’ resistance to pathogen infection, significantly regulates the accumulation of Put and the induction of ADC2 [32]. Here, this study suggests that Put also plays an important positive role in the regulation of cotton fiber cell elongation. Addition of 10 μmol/L Put to the culture medium significantly stimulated cotton fiber cell elongation.

Compared with diploid Asiatic cotton (*G. arboreum*), tetraploid upland cotton (*G. hirsutum*) is a more mature receptor for studying genetic transformation of cotton; therefore, this study focused on the *GhADC2* gene of tetraploid upland cotton. Genome-wide transcriptome profiling of fiber elongation with exogenous Put addition, 3163 DEG were detected, including many fiber elongation-related genes. Ethylene and very long chain fatty acids (such as KCS6) have been reported to stimulate fiber cell elongation [33]. ATP promotes cotton fiber cell elongation, both 1 μmol/L of L-(2-aminoethoxyvinyl)-glycine and 2 μmol/L of acetochlor (2-chloro-N-(ethoxymethyl)-N-(2-ethyl-6-methyl-phenyl)-acetamide reduced fiber length, and application of an ATP biosynthesis inhibitor (oligomycin) significantly inhibited fiber cell elongation, while exogenous addition of 0.1 μmol/L of ethylene or 5 μmol/L of C24:0 effectively negated inhibition of fiber elongation by oligomycin [34]. Cotton *GASL* genes encoding putative gibberellin-regulated proteins [35], Heat Shock proteins *Hsp90* [36], *WRKY transcription factor protein 1* [37], transcription factor *TCP14* [38], *fasciclin-like arabinogalactan protein 2* [39], *DET2* (a steroid 5a-reductase) [40], *receptor-like protein kinase* [37], the R2R3 *MYB* transcription factor [41] are involved in fiber elongation.

In summary, ADC and Put are involved in fiber development of *G. hirsutum* cotton. Expression pattern of *GhADC2* and levels of polyamine are correlated with fiber elongation, especially, the amount of Put. Moreover, exogenous addition of an ADC substrate or Put increased the fiber length. Further, 3163 DEG were detected in genome-wide transcriptome profiling of fiber elongation with exogenous Put addition, including fiber elongation-related genes transcription Factor *MYB109*, *WRKY1*, and *TCP14*.

## Figures and Tables

**Figure 1 genes-13-00784-f001:**
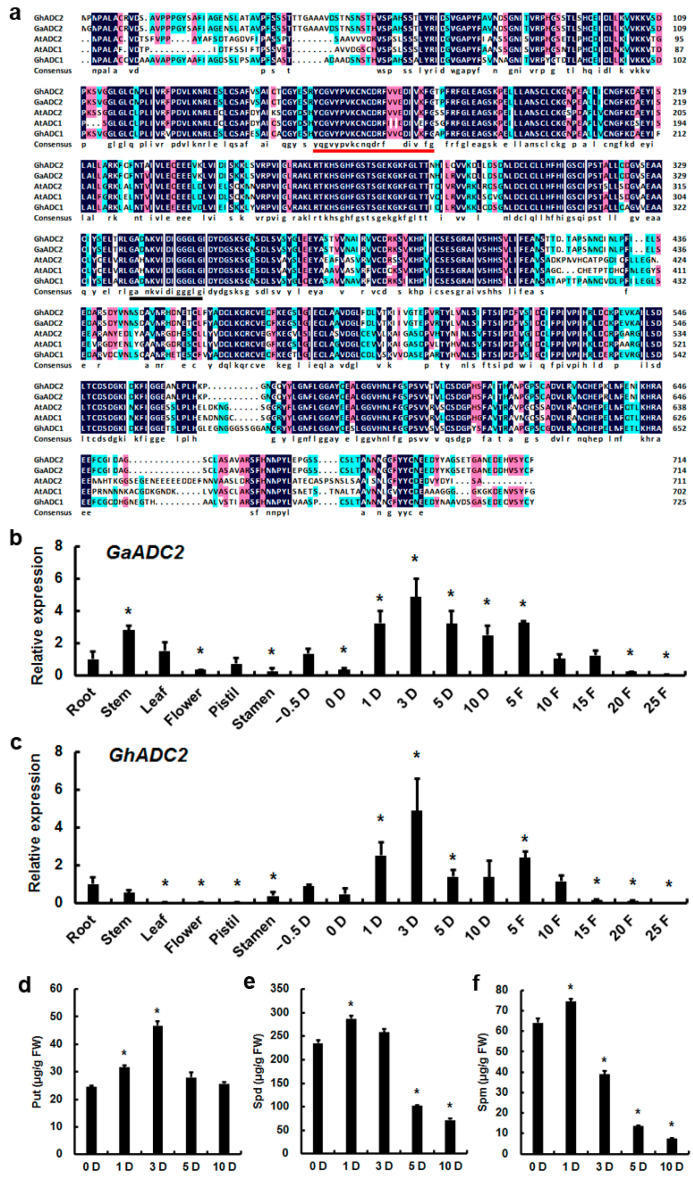
Spatiotemporal expression of *ADC2* and polyamine content in cotton fiber elongation. (**a**) Multiple sequence alignment of GhADC2 and homologous proteins. The red underline is the Orn/DAP/Arg decarboxylasesfamily 2 pyridoxal-P attachment site. The black underline is the substrate recognition signal Orn/DAP/Arg decarboxylases family 2 signature 2. (**b**,**c**) Spatiotemporal expression of *GaADC2* and *GhADC2* in *G. arboreum* (**b**) and *G. hirsutum* (**c**), respectively. D, days post-anthesis. F, days of fiber. Error bars represent the standard deviations of three biological replicates. Asterisks indicate significant difference in transcript abundance compared with the control (ANOVA, Tukey’s test, *p* < 0.05). (**d**) Put content analysis. (**e**) Spd content analysis. Error bars represent the standard deviations of three biological replicates. Asterisks indicate significant difference in transcript abundance compared with the control (ANOVA, Tukey’s test, *p* < 0.05). (**f**) Spm content analysis. FW, fresh weight. Error bars represent the standard deviations of three biological replicates. Asterisks indicate significant difference compared with that of the control 0 DPA (ANOVA, Tukey’s test, *p* < 0.001).

**Figure 2 genes-13-00784-f002:**
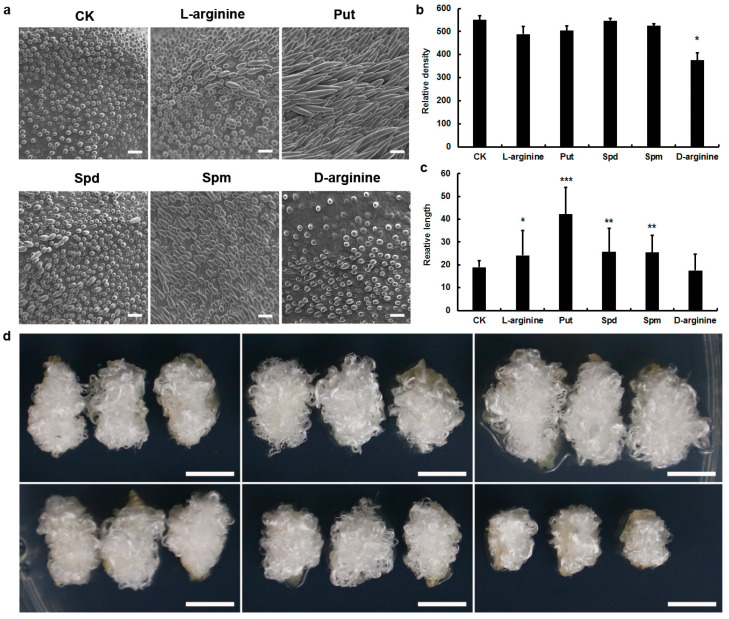
Polyamine (PA) promotes fiber cell elongation. (**a**) Scanning electronic micrographs of ovule surface with addition of 10 μmol/L polyamines, substrate of ADC (L-arginine), or specific Put biosynthesis inhibitor (D-arginine) after 2 d of culture. CK is water control. Bar = 50 μm. (**b**,**c**) Fiber relative density and length with addition of polyamines, L-arginine, or D-arginine. CK is water control. Error bars represent standard deviation (SD) of three biological replicates (ANOVA, Tukey’s test, * *p* < 0.05, ** *p* < 0.01, *** *p* < 0.001). (**d**) Photograph of cotton ovules cultured in vitro for 7 d. The first row is application of 10 μmol/L PBS, L-arginine and Put; second is Spd, Spm, and D-arginine from left to right. Bar = 0.5 cm.

**Figure 3 genes-13-00784-f003:**
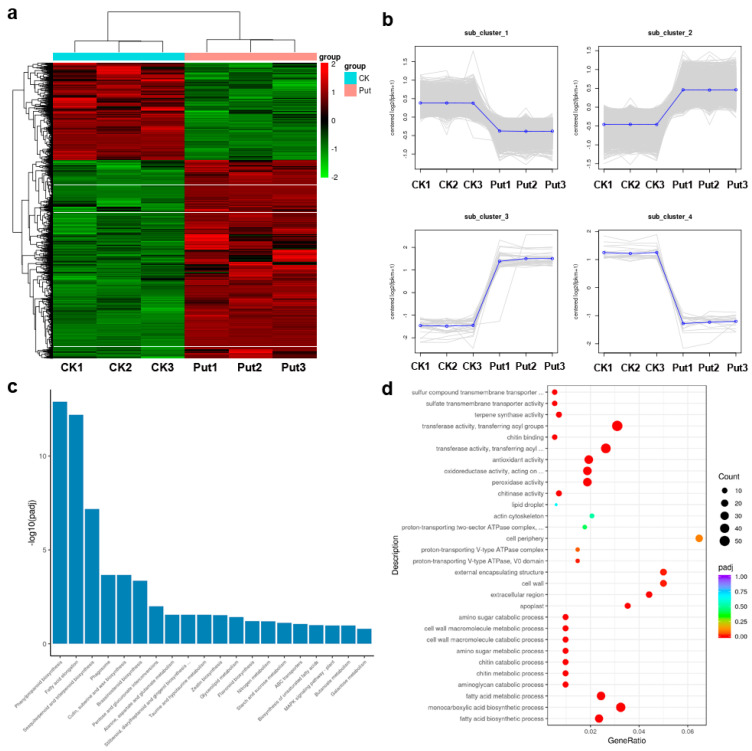
DEG analysis. (**a**) Overall cluster analysis of DEG. FPKM (fragments per kilo base of transcript per million base pairs sequenced) was used to estimate the level of gene expression. The color change from red (highly expressed) to green (low expression) represents the relative expression level value log2 (ratios). (**b**) The H_cluster analysis of DEG resulted in their categorization into four expression pattern types. Each profile presents a gene expression trend. The mean expression levels of all genes in each of the four categories (blue line) are displayed relative to a log2 ratio of 0 (red line). (**c**) Kyoto Encyclopedia of Genes and Genomes (KEGG) pathways analysis of the DEG genes with Put treatment for 2 d then water control. (**d**) GO category enrichment of up-regulated and down-regulated DEG between the Put treatment and water control. The number of genes in each category is equal to the dot size. The dot color represents the *q*-value.

**Figure 4 genes-13-00784-f004:**
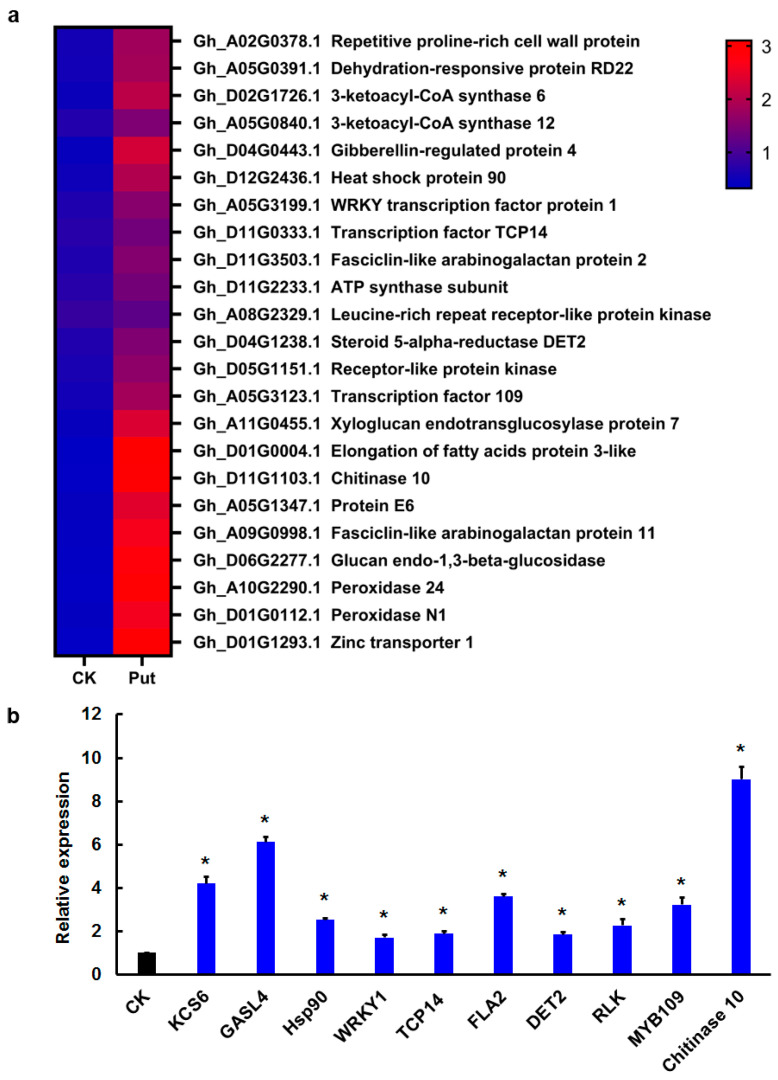
Put regulates the expression of fiber elongation-related genes. (**a**) Transcriptome analysis of DEG in fibers with addition of 10 μmol/L of Put. (**b**) qRT-PCR analysis of genes related to cotton fiber development in 2 DPA fibers of Put treatment. Error bars represent the standard deviations of three biological replicates. Asterisks indicate significant difference in transcript abundance compared with the control (ANOVA, Tukey’s test, *p* < 0.05). *KCS6* (Gene ID, Gh_D02G1726.1), *GASL4* (Gene ID, Gh_D04G0443.1), *Hsp90* (Gene ID, Gh_D12G2436.1), *WRKY1* (Gene ID, Gh_A05G3199.1), *TCP14* (Gene ID, Gh_D11G0333.1), *FLA2* (Gene ID, Gh_D11G3503.1), *DET2* (Gene ID, Gh_D04G1238.1), *RLK* (Gene ID, Gh_D05G1151.1), *MYB109* (Gene ID, Gh_A05G3123.1), *Chitinase 10* (Gene ID, Gh_D11G1103.1).

**Table 1 genes-13-00784-t001:** Primers used in this study.

Primer Name	Forward and Reverse Primers (5’-3’)	Gene ID
GhUBQ7 FW	GAAGGCATTCCACCTGACCAAC	DQ116441
GhUBQ7 RV	CTTGACCTTCTTCTTCTTGTGCTTG	
GhADC2 FW	TTACGACGGTTCAAAATCTG	Gh_A04G1054.1
GhADC2 RV	GCTCTCCCACTCTCGCTACA	
GaADC2 FW	GTTCTCAAATCCCTTCCACG	Cotton_A_08902
GaADC2 RV	GCTCTCCCACTTTCGCTACA	
KCS6 FW	CTCCTGCTATTCATTACATCCC	Gh_D02G1726.1
KCS6 RV	AGGTTTCAGCCCCGTTTT	
GASL4 FW	CTTGTGGCTGCTTTCTTCTTG	Gh_D04G0443.1
GASL4 RV	ACATTGGTAACTCTTGAGGCTTC	
Hsp90 FW	GTGGTATTGGGATGACTAAAGC	Gh_D12G2436.1
Hsp90 RV	GGTGACCGTGAAAGAGCC	
WRKY1 FW	GGCAGATTCGGTGAGCAG	Gh_A05G3199.1
WRKY1 RV	TCAGGAGAGCAAGTGGGC	
TCP14 FW	AGGAAACGAAGACCCGAAC	Gh_D11G0333.1
TCP14 RV	CCGCTACTGCTTGAACCC	
FLA2 FW	CCAAGCATTACTCCCTCTACAC	Gh_D11G3503.1
FLA2 RV	AACCCAACTTTTCCGCCT	
DET2 FW	GCCAACATTTCTACAACCCTAA	Gh_D04G1238.1
DET2 RV	CATTCACCCACATACCAACTACA	
RLK FW	CCTCCACTCTCCTTGCCG	Gh_D05G1151.1
RLK RV	CGACCGCCCTCACCTAAA	
MYB109 FW	AGGGATTATGGGCAATGGAG	Gh_A05G3123.1
MYB109 RV	TTCAAACCTGTTCTGTTGGCTAT	
Chitinase 10 FW	GAAGAGCCCCTCCGTCCA	Gh_D11G1103.1
Chitinase 10 RV	GCACAAGCCCCAAGCGTAT	

**Table 2 genes-13-00784-t002:** RNA-seq mapping and QC data.

Sample	Total_Reads	Total_Map	Unique_Map	Q30	GC_pct
CK1	43,969,370	42,606,649(96.9%)	40,361,736(91.8%)	93.5	43.9
CK2	44,697,902	43,309,165(96.89%)	41,031,266(91.8%)	93.44	43.9
CK3	44,061,224	42,659,853(96.82%)	40,396,630(91.68%)	93.41	43.97
Put1	42,464,198	41,119,927(96.83%)	38,973,126(91.78%)	93.42	44.04
Put2	38,983,794	37,788,560(96.93%)	35,866,850(92.0%)	93.59	44.01
Put3	39,482,912	38,276,222(96.94%)	36,325,782(92.0%)	93.42	43.93

## Data Availability

Not applicable.

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
