# Peer review of "Arginine Decarboxylase Gene ADC2 Regulates Fiber Elongation in Cotton"

_genes, 2022, doi:10.3390/genes13050784_

Round 1

Reviewer 1 Report

The manuscript “Arginine decarboxylase gene ADC2 regulates fiber elongation in cotton” fits the journal’s scope. The authors present the results regarding the arginine decarboxylase and putrescine involvement in the development and fiber elongation of G. hirsutum. The design of the study is and the methodology are described in sufficient detail, and the results are presented adequately. The manuscript is well written, clear, and the quality of the results’ presentation is high. The manuscript need some minor corrections (please see them below).

The novelty of the research should be highlighted.

Measurements of polyamine: please indicate the sample preparation and the range of concentration for each standard curve.

Author Response

Response to Reviewer 1 Comments

The manuscript “Arginine decarboxylase gene ADC2 regulates fiber elongation in cotton” fits the journal’s scope. The authors present the results regarding the arginine decarboxylase and putrescine involvement in the development and fiber elongation of G. hirsutum. The design of the study is and the methodology are described in sufficient detail, and the results are presented adequately. The manuscript is well written, clear, and the quality of the results’ presentation is high. The manuscript need some minor corrections (please see them below).

Point 1: The novelty of the research should be highlighted.

Response 1: Your suggestions are greatly appreciated and have been adopted. The novelty of the research has been highlighted in the ABSTRACT.

Point 2: Measurements of polyamine: please indicate the sample preparation and the range of concentration for each standard curve.

Response 2: Thank you for giving us constructive suggestions. We’ve adopted your suggestions and indicated the sample preparation and the range of concentration for each standard curve as follows:

The polyamine content was analyzed via HPLC (Shimadzu LC-20A) on an apparatus equipped with an Agilent reversed-phase C18 column (particle size, 4 μm; column, 3.9 mm × 150 mm), and was detected at 230 nm with a method slightly modified from that previously described [1]. Polyamine standards or 1 g fresh plant tissue was ground with 5 mL perchloric acid in ice bath, extracted for 30 min at 4 °C, centrifuged at 12000 rpm for 30 min at 4 °C, 2 mL supernatant was added with 7 μL benzoyl chloride and 1 mL NaOH for derivatization, followed by incubate for 20 min at 37 °C. Then 2 mL diethyl ether and 2 mL saturated NaCl was added, centrifuged at 3700 rpm for 5 min. After vacuum drying for 10 min, add 500 μL 60% methanol to dissolve and wait for measurement. Vortex for 5 min, then filter through 0.45 μm organic membrane. The injection volume of 10 μL was determined by HPLC. The standard curves of Put, Spd, and Spm were made by external standard method, and the content of samples was analyzed. The retention time of each polyamine in the mixed standard product was determined first, and then the retention time of Put, Spd, and Spm in the sample was determined due to the different number of amino groups of the three polyamines. Put was the first to be separated, followed by Spd and Spm. The standard curve of Put exhibits y=1.0885x-4.0307, R2=0.9996, retention time 7.6 min, the range of concentration for Put standard curve was from 10 μM to 500 μM. The standard curve of Spd reveals y=1.0721x-1.0879, R2=0.9999, retention time 12 min, the range of concentration for Spd standard curve was from 200 μM to 2000 μM. The standard curve of Spm shows y=0.8354x-1.912, R2=0.9998, and retention time 15.1 min, the range of concentration for Spm standard curve was from 10 μM to 1000 μM.

Reviewer 2 Report

Introduction: Maybe it can be  improved, since the central idea of what the research will cover in terms of making clear if ADC is a key rate-limiting enzyme and Put, Spd, and Spm as components of polyamine metabolic pathway is not so easy to follow. 

62 the initials ODC have not been written in full

68 members

81 (in China)

88 to 91 Say that spd, spm and put are measured by HPLC and compared to gene expression on different tissues

117-118 The gene expression analysis was performed with tissue of three different samples (biological replicates) and three repetitions per sample (technical replicates)?

121 root or water control of 2 d ovule samples?

164 Is it genetic transformation success relevant here? Better to talk about at the discussion?

166 which may be designated GhADC2

221 deleteexhibited’

223 involved?

Figure 2: just nice

254 93.41% to 93.59% of the Q30 sequence quality data was obtained?

268 , 270 can you use another verb instead of ‘having’

273-275   up and down regulated lists can be separated? Or are there up and down regulated genes within these pathways?

277 collected. In

281-3 are elucidated?

315 H2O2

336 G. hirsutum cotton

Author Response

Response to Reviewer 2 Comments

Point 1: Introduction: Maybe it can be improved, since the central idea of what the research will cover in terms of making clear if ADC is a key rate-limiting enzyme and Put, Spd, and Spm as components of polyamine metabolic pathway is not so easy to follow.

Response 1: Thank you for giving us constructive suggestions. We’ve adopted your suggestions and therefore improved the introduction as follows: Put, spermidine (Spd), and spermine (Spm), as components of polyamine metabolic pathway, are the most common di- and polyamines in seed plants[2].

ADC is a key rate-limiting enzyme in the polyamine metabolic pathway, catalyzes the synthesis of Put from L-arginine [3].

Point 2: 62 the initials ODC have not been written in full

Response 2: The suggestion is greatly appreciated. We have revised as you suggested: There is no definite evidence of ornithine decarboxylase (ODC) and its encoding gene in cruciferous plants such as Arabidopsis thaliana [1].

Point 3: 68 members

Response 3: The suggestion is greatly appreciated. We have revised as you suggested: In plants, the differential expression patterns of ADC gene members is observed.

Point 4: 81 (in China)

Response 4: The suggestion is greatly appreciated. We have revised as you suggested: In our previous research, the ADC gene (GhADC1, GenBank, KC851856) has been cloned in Gossypium hirsutum (G. hirsutum), where it revealed variable expression under Verticillium wilt (In China).

Point 5: 88 to 91 Say that spd, spm and put are measured by HPLC and compared to gene expression on different tissues

Response 5: The suggestion is greatly appreciated. We have revised as you suggested: Here, we demonstrated that a cotton ADC gene (GhADC2) promotes fibre elongation. Spd, spm and put are measured by HPLC and compared to gene expression on different tissues.

Point 6: 117-118 The gene expression analysis was performed with tissue of three different samples (biological replicates) and three repetitions per sample (technical replicates)?

Response 6: The suggestion is greatly appreciated. We have revised as you suggested: The gene expression analysis was performed with tissue of three different samples (biological replicates) and three repetitions per sample (technical replicates).

Point 7: 121 root or water control of 2 d ovule samples?

Response 7: The suggestion is greatly appreciated. We have revised as you suggested: The results were normalized to the expression level of UBQ7 (GenBank: DQ116441) and relative to the root (in Fig. 1b) or 2 d ovule samples without Put treatment (in Fig. 4b).

Point 8: 164 Is it genetic transformation success relevant here? Better to talk about at the discussion?

Response 8: The suggestion is greatly appreciated. We have added this information in discussion: Compared with diploid Asiatic cotton (G. arboreum), tetraploid upland cotton (G. hirsutum) is a more mature receptor for studying genetic transformation of cotton, therefore, this study focused on the GhADC2 gene of tetraploid upland cotton.

Point 9: 166 which may be designated GhADC2

Response 9: The suggestion is greatly appreciated. We have revised as you suggested: which may be defined as GhADC2 (Gene ID, Gh_A04G1054.1).

Point 10: 221 delete ‘exhibited’

Response 10: The suggestion is greatly appreciated. We have deleted ‘exhibited’.

Point 11: 223 involved?

Response 11: The suggestion is greatly appreciated. We have revised as you suggested: Overall, the trends of up-regulated Put content and the expression pattern of GhADC2 are similar during fiber elongation, suggesting that ADC-mediated Put level involved in fiber elongation.

Point 12: Figure 2: just nice

Response 12: The suggestion is greatly appreciated. Thank you.

Point 13: 254 93.41% to 93.59% of the Q30 sequence quality data was obtained?

Response 13: The suggestion is greatly appreciated. We have revised as you suggested: Meanwhile, 93.41% to 93.59% of the Q30 sequence quality data was obtained.

Point 14: 268 , 270 can you use another verb instead of ‘having’

Response 14: The suggestion is greatly appreciated. We have revised as you suggested: Type I, covering 1041 genes, included clusters 1 and 4, which were down-regulated in ovules with addition of 10 μmol/L Put after 2 d of culture, and Type II, involving 2122 genes, included clusters 2 and 3, which were up-regulated.

Point 15: 273-275 up and down regulated lists can be separated? Or are there up and down regulated genes within these pathways?

Response 15: The suggestion is greatly appreciated. We have revised as you suggested: Based on our annotation of transcripts, phenylpropanoid biosynthesis and fatty acid elongation pathways were up-regulated, whereas sesquiterpenoid and triterpenoid biosynthesis, brassinosteroid biosynthesis and MAPK signaling pathways down-regulated in ovule samples after 2 d of culture with Put treatment.

Point 16: 277 collected. In

Response 16: The suggestion is greatly appreciated. We have revised as you suggested: Next, GO enrichment analysis of the DEG was conducted, and then 86 pathways were collected. In the DEG, genes were enriched mainly in fatty acid biosynthetic process (GO:0006633),

Point 17: 281-3 are elucidated?

Response 17: The suggestion is greatly appreciated. We have revised as you suggested: Thus, these data indicate that the functional enrichment of biological pathways may participate in the Put-mediated signal transduction.

Point 18: 315 H2O2

Response 18: The suggestion is greatly appreciated. We have revised as you suggested: accompanied by adaptation of activities of antioxidant enzymes and the level of H2O2, in Korean ginseng sprouts.

Point 19: 336 G. hirsutum cotton

Response 19: The suggestion is greatly appreciated. We have revised as you suggested: In summary, ADC and Put are involved in fiber development of G. hirsutum cotton.

Kind regards,

Huijuan Mo

State Key Laboratory of Cotton Biology, Institute of Cotton Research of the Chinese Academy of Agricultural Sciences, Anyang, China
